# A Computer-Aided Drug Design Approach to Predict Marine Drug-Like Leads for SARS-CoV-2 Main Protease Inhibition

**DOI:** 10.3390/md18120633

**Published:** 2020-12-10

**Authors:** Susana P. Gaudêncio, Florbela Pereira

**Affiliations:** 1Blue Biotechnology & Biomedicine Lab, UCIBIO—Applied Biomolecular Sciences Unit, Department of Chemistry, Faculty of Sciences and Technology, NOVA University of Lisbon, 2829-516 Caparica, Portugal; s.gaudencio@fct.unl.pt; 2LAQV, Department of Chemistry, Faculty for Sciences and Technology, NOVA University of Lisbon, 2829-516 Caparica, Portugal

**Keywords:** marine natural products (MNPs), drug discovery, actinomycetes, quantitative structure–activity relationship (QSAR), machine learning (ML) techniques, molecular docking, virtual screening, severe acute respiratory syndrome coronavirus 2 (SARS-CoV-2), main protease enzyme (M^pro^)

## Abstract

The investigation of marine natural products (MNPs) as key resources for the discovery of drugs to mitigate the COVID-19 pandemic is a developing field. In this work, computer-aided drug design (CADD) approaches comprising ligand- and structure-based methods were explored for predicting SARS-CoV-2 main protease (M^pro^) inhibitors. The CADD ligand-based method used a quantitative structure–activity relationship (QSAR) classification model that was built using 5276 organic molecules extracted from the ChEMBL database with SARS-CoV-2 screening data. The best model achieved an overall predictive accuracy of up to 67% for an external and internal validation using test and training sets. Moreover, based on the best QSAR model, a virtual screening campaign was carried out using 11,162 MNPs retrieved from the Reaxys^®^ database, 7 in-house MNPs obtained from marine-derived actinomycetes by the team, and 14 MNPs that are currently in the clinical pipeline. All the MNPs from the virtual screening libraries that were predicted as belonging to class A were selected for the CADD structure-based method. In the CADD structure-based approach, the 494 MNPs selected by the QSAR approach were screened by molecular docking against M^pro^ enzyme. A list of virtual screening hits comprising fifteen MNPs was assented by establishing several limits in this CADD approach, and five MNPs were proposed as the most promising marine drug-like leads as SARS-CoV-2 M^pro^ inhibitors, a benzo[f]pyrano[4,3-b]chromene, notoamide I, emindole SB beta-mannoside, and two bromoindole derivatives.

## 1. Introduction

To date, over 68.2 million people worldwide have been infected by coronavirus disease 2019 (COVID-19), resulting in over 1.5 million deaths caused by severe acute respiratory syndrome (SARS-CoV-2) and in addition, severe economic costs, national health systems crises, and massive unemployment [1,2]. Despite the enormous human and financial efforts, there is still no appropriate treatment or prevention for COVID-19 and the number of deaths keeps increasing, which makes the discovery of drugs for infection treatment of foremost importance and an emergency. New and efficient therapeutic agents will relieve social, economic, and management burdens, improving patients’ quality of life, and reducing hospitals pressure due to ineffective/long hospitalizations.

The vital role played by the protease enzyme (M^pro^) in polyprotein processing and virus maturation makes M^pro^ a target for the design of antiviral drugs towards COVID-19 treatment [3,4,5,6,7,8,9]. The crystal structure of M^pro^ at 2.16 Å in complex with a covalent inhibitor was reported by Zihe Rao (Protein Data Bank, PDB, ID 6LU7, https://www.rcsb.org/3d-view/6LU7/1) [3], allowing the development of structure-based computer-aided drug design (CADD) approaches, such as molecular docking [10,11]. Other approaches explore other targets such as the SARS-CoV-2 spike protein that binds to the human angiotensin-converting enzyme 2 (ACE2) receptor [12,13].

Oceans have yielded sixteen approved drugs so far, and dozens of MNPs for clinical trials, the great majority of which are intended for cancer treatment [14]. However, as far as discovering MNPs are concerned to mitigate antiviral infections, it is mostly an unexplored field as it was highlighted by Riccio et al. in a recent review [15]. Ara A^®^ is the only marine approved antiviral drug for herpes simplex. In clinical trials, there is Griffithsin^TN^, a lectin extracted from a red algae and a macrolide naturally produced by marine bacterial symbionts of the bryozoan Bugula neritina, the previously approved anti-cancer Bryostatin^®^, both for human immunodeficiency virus (HIV) [15]. Recently, PharmaMar publicly announced that the anti-tumor drug Aplidin^®^ (plitidepsin) revealed higher effect for coronavirus treatment than Ivermictin^®^, obtained from soil-derived actinomycete, and synthetic Remdesivir^®^, beginning Phase II clinical trials for COVID-19 treatment.

In the quest to discover solutions for the COVID-19 pandemic from the marine environment, researchers are redirecting their studies to this topic using structure-based CADD methodologies. Gentile et al. suggested 17 lead-like inhibitors of the M^pro^, belonging to the compound class phlorotannins, by screening a MNP library of 14,064 compounds isolated from *Sargassum* spp. brown alga, through the hyphenated pharmacophore model and molecular docking approaches to predict inhibitors of M^pro^ obtained from PDB (ID 6LU7) [5]. Khan et al. docked to the M^pro^ target from PDB (ID 6MO3), five MNPs from the PubChem database, two MNPs isolated from sponges of the species *Petrosia* and the family Aplysinidae, and one MNP from the soft coral *Pterogorgia citrina*, among these, the (11*R*)-11-epi-Fistularin-3 of the Aplysinidae sponge was predicted as lead-like inhibitor against SARS-CoV-2 [6]. 

Several studies have been reported with the development of ligand-based CADD approaches for the discovery of inhibitors against SARS-CoV-2 [16,17,18]. Ghosh et al. reported the development of several Monte Carlo optimization-based, quantitative structure–activity relationship (QSAR) models with a diverse dataset comprising 88 compounds with SARS-CoV-2 M^pro^ assay from the ChEMBL database and the best model was used for virtual screening of 60 NPs from recent publications [16]. Using the virtual screening, the authors proposed thirteen NPs as the most potent virtual hits for M^pro^ inhibition including one lignan, eleven flavonoids, and one pentacyclic triterpenoid. The authors also suggested that heterocyclic scaffolds such as diazole, furan, and pyridine have a positive contribution, while thiophen, thiazole, and pyrimidine appear to have a negative contribution to the M^pro^ inhibition [16]. Another study correlated the activity against SARS-CoV-2 M^pro^ with the presence of a different *N*-heterocyclic scaffold, such as a pyridone ring [19].

Despite the fact that the interactions between marine viral and bacterial species are under investigation, in the marine environment, the number of viruses is 10 to 25-fold higher than bacteria, which suggests that marine bacteria have evolved to co-exist with numerous viruses producing MNPs with a broad-range of antiviral activities to compete for survival [20,21,22]. Our group has extensive experience in both marine-derived actinomycetes [23,24,25] and MNP modeling and virtual screening [26,27,28,29] being compelled to provide marine drug-leads to feed the NHS clinical trials for COVID-19 infection treatment and the pharmaceutical pipelines. Herein, we report a comprehensive computational modeling for the prediction of SARS-CoV-2 M^pro^ inhibitors from three MNP libraries, by employing structure- and ligand-based CADD methodologies. MNPs libraries comprised: (1) 11,162 MNP retrieved from the Reaxys^®^ database, (2) 7 in-house MNPs obtained by the team from marine-derived actinomycetes, and (3) 14 MNPs from MNPs clinical pipeline (eight approved drugs and six MNPs in Phase II and III of clinical trials). All the MNPs from the virtual screening libraries that were predicted as belonging to the class A, were selected to proceed to the CADD structure-based method. Where 494 MNPs selected by QSAR approach were screened by molecular docking against M^pro^ enzyme. In this CADD approach, a list of virtual screening hits comprising fifteen MNPs was assented on the basis of some established limits, such as: confidence value (**3**), probability of being active against SARS-CoV-2 in the best model, prediction of the affinity between the M^pro^ of the selected MNPs through molecular docking. Five MNPs, a benzo[f]pyrano[4,3-b]chromene, notoamide I, emindole SB beta-mannoside, and two bromoindole derivatives were proposed as the most promise marine drug-like leads as SARS-CoV-2 M^pro^ inhibitors.

## 2. Results and Discussion

### 2.1. Chemical Space of the SARS-CoV-2 Model

The whole data set of 5272 organic molecules from the ChEMBL database with SARS-CoV-2 screening data (antiviral activity determined as inhibition of SARS-CoV-2 induced cytotoxicity of Caco-2 cells) was randomly divided into a training set of 3499 molecules (comprising 302 molecules from class A with inhibition % ≥50%, 265 molecules from class B with 50% > inhibition % ≥ 30%, and 2932 molecules from class C with inhibition % < 30%), a test set of 1533 molecules (comprising 145 molecules from class A with inhibition % ≥ 50%, 99 molecules from class B with 50% > inhibition % ≥ 30%, and 1288 molecules from class C with inhibition % < 30%), and an additional test set, test 2 set, of 241 molecules, which were used for the development and external validation of the QSAR classification models, respectively. The whole data set was clustered into ten structural classes or scaffold types (I–X) using the ward tool in JChem. The ten structural clusters are represented in Table 1 along with their average of molecular weight (MW), the octanol-water partition coefficient estimation (AlogP), and activity class counts.

The percentage of molecules belonging to class A, i.e., active against SARS-CoV-2, in the training set is relatively low, 9%, and these molecules are distributed within the ten structural groups (I–X), with percentages ranging from 3–13%. There are three structural clusters (III, VII, and VIII) in which class A is less represented compared to its representation in the training set, with percentages between 3–5%. Moreover, in the other seven structural clusters, class A has a representation equal to or higher than that obtained for class A in the training set, with percentages between 9–13%. The well-known, Lipinski rule, informs mainly if a molecule is more likely to be an orally administrated active drug and if it is easily absorbed by the body. One of the most important parameters is the LogP, which is highly correlated with lipophilicity, thus, highly lipophilic molecules are often discontinued from drug development and are frequently related to toxicity issues [30]. To explore the chemical and biological diversity of the training set, the active, intermediate, and inactive molecules against SARS-CoV-2 in the training set were analyzed, according to the ten structural clusters, using MW and ALogP. The analysis of these data indicates that the active, intermediate, and inactive molecules against SARS-CoV-2 in the training set are distributed over a wide range of MW (i.e., 84–4187 Da) and ALogP (i.e., −8.9–17.85). Interestingly, approximately 84% of the molecules present in the training set have a MW bellow 500 Da. This MW interval contains approximately 78%, 82%, and 85% of all active, intermediate, and inactive molecules against SARS-CoV-2 in the training set, respectively. In spite of this, using this rule (MW < 500 Da), it is possible to discriminate actives molecules in relation to intermediate and inactive molecules in three structural clusters, namely in the clusters III, IV, and X, which comprises 80%, 82%, and 86% of inactive molecules, respectively, when compared to 78% of active molecules in the overall training set. In addition, more than 68%, 82%, and 86% of the active, intermediate, and inactive molecules against SARS-CoV-2 in the training set have an ALogP that is lower than 5, respectively. In the same way, using the ALogP < 5 rule, it is possible to prioritize active molecules in relation to the other molecules in three clusters, namely in the clusters: VII, VIII and IX, which comprises 85%, 81%, and 79% of active molecules when compared to 68% of active molecules in the overall training set, respectively.

### 2.2. QSAR Classification Modeling

Two extensive sets of descriptors were explored, one with 6 different types of fingerprints (FPs) with different sizes (166 MACCS, MACCS keys; 307 Substructure, presence and count Sub and SubC, respectively; 881 PubChem fingerprints; 1024 CDK, circular fingerprints; and 1024 CDKExt, extended circular fingerprints with additional bits describing ring features) and another with a total of 1442 1D&2D descriptors (including electronic, topological, and constitutional descriptors). The FPs and the molecular descriptors were calculated by PaDEL-Descriptor [31]. Random forests (RF) [32] machine learning (ML) technique was used for building SARS-CoV-2 activity prediction classification models, and the performance of the models was successfully evaluated by internal validation (out-of-bag, OOB, estimation on the training set), Table 2.

In Table 2, were highlighted in bold the three selected models, MACCS model—best model built with sets of fragment FPs (MACCS, Sub, SubC, and PubChem), ExtCDK model—best model built with sets of circular FPs (CDK and ExtCDK), and 1D&2D descriptors in accordance with Q and MCC parameters. For the two best models for the training set, ExtCDK, and 1D&2D (Table 2), the descriptor selection was evaluated based on the importance assigned by the RF model with the R program—Table 3. For the MACCS model, descriptors were not selected since these comprised only 166 FPs.

The best models for each set of ExtCDK FPs and 1D&2D descriptors were accomplished with the RF algorithm using the 150 selected FPs and descriptors, respectively, (the best models were highlighted in bold—Table 3) for the training set. Majority voting predictions (consensus) obtained by ExtCDK, 1D&2D, and MACCS models (consensus model, CM) further improved the results with a Q and MCC of 0.68 and 0.31 for the training set, respectively, Table 4. The results obtained by the CM for the test set in accordance with the ten structural categories (I–X), which were set up using the ward tool in JChem, are shown in Table 5.

In general, the predictions obtained for the structural clusters are better than those obtained for the overall test set simultaneously taking into account the Q and MCC values, except for Clusters IV–V and IX–X (bold highlighted in Table 5). An improvement in the CM model prediction accuracies (Q = 0.67–0.72 and MCC = 0.25–0.38) was achieved for the other clusters (Clusters I–III and VI–VIII) of the test set, when compared with the prediction accuracy obtained for all the molecules of the test set (Q = 0.67 and MCC = 0.19). For the clusters IV–V and IX–X inferior prediction accuracies were obtained, Q = 0.58–0.68 and MCC = 0.05–0.21. Interestingly, the best achieved predictions for structural clusters I and VI are related to the best performance obtained for the class A (active) prediction with SE values of 0.71 and 0.67, respectively, compared to the SE value of 0.48 for all test sets. Furthermore, an additional predicting criterion can be generated according to the vote count of each of the models (i.e., MACCS, ExtCDK, and 1D&2D). Thus, if classes A, B, or C are predicted by the three models, these have the maximum confidence value (3), by two models (2), and by only one model (1), in this case, the CM prediction is obtained by the prediction of the best model, ExtCDK. An additional parameter, probability of being of class A (Prob_A) or probability of being active, was assigned by the RF algorithm and it can be used as a predicting criterion. For instance, the percentage of TA with the maximum confidence value (3) was 57%, which compares with the percentages of 42% and 43% obtained for false A, FA_B (false A which was B) and FA_C (false A which was C), respectively. If the criterion Prob_A ≥ 0.5 was added for the percentage of TA, the percentage of 33% was obtained and it compares with percentages of 13% and 15% obtained for FA_B and FA_C, respectively.

There are thirty-seven molecules from test set 2 that were predicted as being for class A, from those fourteen, were predicted with the maximum confidence value (3) and only five were predicted with Prob_A ≥ 0.5. Chemical structures of the five molecules in test set 2 that were predicted as belonging to class A with the confidence value (3) and Prob_A ≥ 0.5 are described in Figure 1.

NCATS released, via OpenData Portal (https://opendata.ncats.nih.gov/covid19/assays) the quantitative HTS data on drugs approved for clinical use tested in the SARS-CoV-2 cytopathic effect (CPE) assay. We used the chemical structures and data of 3957 molecules in SMILES format of this SARS-CoV-2 CPE evaluation from the work of Alves et al. [17]. From those 3957 molecules, there are 1401 molecules that were used to build (910 molecules in the training set) and validate (409 and 82 molecules in the test and test 2 sets, respectively) the SARS-CoV-2 QSAR model. Only the KRCA-0008 molecule in Figure 1, which was proposed as the most promising active molecules against SARS-CoV-2 for test set 2, was tested by NCATS and had no activity. However, there is a proposed hit of the test 2 set, quizartinib (Figure 2), predicted with the maximum confidence value (3) but with Prob_A < 0.5 by the QSAR model that was experimentally validated by the SARS-CoV-2 CPE assay as active with AC_50_ of 2.51 µM.

All six molecules predicted to be the most promising against SARS-CoV-2 from the test set 2 in Figure 1 and Figure 2 are *N*-heterocyclic molecules, four of which have a pyrimidine scaffold (BIBU-1361, CHEMBL214796, NGD-94-1, KRCA-0008), one, OXA-06, has a pyrrolo[2¨C3-b]pyridine scaffold, and quizartinib has a benzo[d]imidazo[2,1-b]thiazole scaffold. Better predictions compared to those obtained for the test set (Table 4) were obtained taking into account the results of the SARS-CoV-2 CPE assay for the 82 molecules of the test set 2 and the values of 0.21, 0.94, and 0.74 for sensitivity, specificity C, and general predictive precision, respectively. Likewise, the other 2556 molecules of the NCATS data were predicted by the CM model, obtaining 0.01, 0.01, 0.98, and 0.91 for sensitivity, specificity B, specificity C, and general predictive precision, respectively. Only one molecule, vorapaxar (Figure 3), was predicted by the QSAR model as true A.

### 2.3. Analysis of MACCS FPs and 1D&2D Descriptors Identified as Relevant for Modeling the Antiviral Activity against SARS-CoV-2

A comparison of the best twenty FPs (i.e., MACCS keys) and descriptors (i.e., 1D_2D) selected by descriptor importance of RF used to build the QSAR classification models is comprised in Table 2 and these were analyzed and presented in Figure 4.

Of the twenty most important MACCS FPs for modeling the antiviral activity against SARS-CoV-2 in the set, there are more MACCS FPs (i.e., eighteen MACCS key) that are relevant in discriminating the class C than the classes A or B. For example, only the 11th, which codifies the presence of *N*-heterocyclic scaffold, and the 19th, which encodes the existence of a 4-membered ring, MACCS FPs out of the twenty most important FPs of MACCs are more relevant in discriminating the A and B classes, respectively. The importance of the *N*-heterocyclic scaffold in class A discrimination has already been highlighted in the proposed list of the six most promising molecules as antiviral against SARS-CoV-2 for test set 2, namely four molecules containing a pyrimidine scaffold, a molecule with a pyrrolo[2,3-b]pyridine scaffold, and a benzo[d]imidazo[2,1-b]thiazole scaffold. In the same way, of the twenty most important descriptors for modeling the antiviral activity against SARS-CoV-2, there are more 1D&2D descriptors that are relevant in discriminating the class C than the classes A or B (i.e., eighteen descriptors) in the set. There are two 1D&2D descriptors that are more relevant in discriminating the class A than the others classes in the set of the twenty most important 1D&2D descriptors for modelling the antiviral activity against SARS-CoV-2, Figure 4, both are topological descriptors, VR3_Dzp (logarithmic Randic-like eigenvector-based index from Barysz matrix/weighted by polarizabilities) and piPC4 (conventional bond order ID number of order 4, log scale). Despite the two most relevant 1D&2D descriptors for class B, an autocorrelation (ATSC0e, centered Broto-Moreau autocorrelation—lag 0/weighted by Sanderson electronegativities) and a topological (ETA_dEpsilon_A, a measure of contribution of unsaturation and electronegative atom count) descriptors, these are more relevant in class C discrimination.

### 2.4. Application of the In Silico Anti-Viral Model Against SARS-CoV-2 in Virtual Screening

A virtual screening campaign was carried out to search for new lead-like inhibitors against SARS-CoV-2. The best model, the CM, was selected for the virtual screening procedure using 11,162 MNPs retrieved from the Reaxys^®^ database, 7 in-house MNPs obtained by the team from marine-derived actinomycetes, and 14 MNPs in the pharmaceutical pipeline (eight approved drugs and six MNPs in Phase II and III of clinical trials). All the MNPs from the virtual screening libraries that were predicted as belonging to the class A, were selected to the CADD structure-based approach. In the CADD structure-based method, the 494 MNPs selected by QSAR classification approach were screened by molecular docking against M^pro^ enzyme.

### 2.5. Molecular Docking Against M^pro^ Enzyme

The 494 MNPs from the three MNPs libraries (491, 1, and 2 from MNPs from Reaxys^®^ database, in-house MNPs, MNPs pharmaceutical pipeline, respectively) selected by QSAR classification approach were screened by molecular docking against M^pro^ enzyme (PDB ID: 6LU7) [3]. The antiviral drugs nelfinavir and lopinavir were used as positive controls and allicin was used as negative control in molecular docking experiments [7]. A list of virtual screening hits comprising thirteen MNPs was assented on the basis of some limits established in this CADD ligand- and structure-based approach, such as: confidence value (3) and probability of being active against SARS-CoV-2 in the best model and prediction of the affinity between the M^pro^ and the selected MNPs through molecular docking (ΔG_B_ ≤ −8 kcal/mol). The Autodock Vina software [33] (http://vina.scripps.edu/) was used to perform the flexible virtual screening of the 494 MNPs from the three MNPs libraries to find the most favorable binding interactions, and the calculated free binding energies by the two sets of search space coordinates were reported in Table 6 for the 15 MNPs selected, one from in-house MNPs (hydroxydebromomarinone), two from MNPs clinical trials (nelarabine and fludarabine), and the positive (nelfinavir and lopinavir) and the negative (allicin) controls.

To prioritize the best marine drug-like leads as SARS-CoV-2 M^pro^ inhibitors from the list of fifteen selected MNPs by QSAR model against SARS-CoV-2 and molecular docking of M^pro^ enzyme, the absorption, distribution, metabolism, excretion, and toxicity (ADMET) properties were predicted via in silico methods using the pKCSM software (http://biosig.unimelb.edu.au/pkcsm/) [34]. Five MNPs, a benzo[f]pyrano[4,3-b]chromene (Reaxys ID 7450892), notoamide I (Reaxys ID 19384758), emindole SB beta-mannoside (Reaxys ID 26845562), and two bromoindole derivatives (Reaxys IDs 10714788 and 10720065) were proposed as marine drug-like leads as SARS-CoV-2 M^pro^ inhibitors. In accordance with Lipinski rule of five, 3 molecules failure in only one rule (one from indoloditerpene and two bromoindole derivatives). Regarding the desirable ADMET properties, no AMES toxicity, hERG I inhibition, or skin sensitization was predicted by the first three hits, and no hERG I inhibition, hepatotoxicity, or skin sensitization by the other two hits, Table 6. The prediction of ADMET properties of the fifteen selected MNPs by QSAR model against SARS-CoV-2 and molecular docking of M^pro^ enzyme was presented in Appendix A, in the Appendix A. In Figure 5, the interaction profiles of the best-docked poses for the five lead-like SARS-CoV-2 M^pro^ inhibitors were represented.

## 3. Materials and Methods

### 3.1. Data Sets and Selection of Training, Test, Test 2 Sets

In total, 5466 organic molecules were extracted from the ChEMBL (https://www.ebi.ac.uk/chembl/) database [35], searching by antiviral activity determined as inhibition of SARS-CoV-2 induced cytotoxicity of Caco-2 cells. Their chemical structures were saved in the simplified molecular input line entry specification (SMILES) data format. The Caco-2 cell line is widely used with in vitro assays to predict the absorption rate of lead drug molecules across the intestinal epithelial cell barrier and it is extensively used to study infection by SARS-CoV and can be used for SARS-CoV-2 infection [36]. The antiviral activity was classified using three activity classes: (A)—inhibition % ≥ 50%; (B)—50% > inhibition % ≥ 30%; (C)—inhibition % < 30%. After calculating the molecular descriptors and the fingerprints, the final data set comprises 5273 organic molecules, the molecules that showed errors in the calculation were removed. From these 447 molecules are recorded as active (class A), 364 as intermediate active (class B), 4220 as inactive (class C), and 241 without defined recorded (label as “*Outside typical range*”). The data set was randomly divided into a training set of 3499 molecules (class A: 302 molecules, class B: 265 molecules, and class C: 2932 molecules), and a test set of 1533 molecules (class A: 145 molecules, class B: 99 molecules, and class C: 1288 molecules), a partition of approximately 56:44 for the training and test sets. The test 2 set comprises the 241 molecules without activity recorded. The SMILES strings of the data set, the corresponding experimental, and the predicted activities are available as Appendix A.

The virtual library comprises 11,162 MNPs set retrieved from the Reaxys^®^ database (Elsevier Information Systems GmbH)) in the MDL SDF data format, 7 in-house MNPs set obtained by the team from marine-derived actinomycetes, and 14 MNPs from the pharmaceutical pipeline set (eight approved drugs and six MNPs in Phase II and III of clinical trials). The SMILES strings of the in-house MNPs and MNPs clinical trials sets, the corresponding predicted activities are available as Appendix A.

### 3.2. Calculation of Molecular Descriptors and Fingerprints

JChem Standardizer tool version 5.7.13.0 (ChemAxon Ltd., Budapest, Hungary) was used to standardize the molecular structures of all data sets by normalizing tautomeric and mesomeric groups and by removing small disconnected fragments. Empirical molecular fingerprints (FPs) and 1D&2D molecular descriptors were calculated by PaDEL-Descriptor version 2.21 (http://www.yapcwsoft.com/dd/padeldescriptor/) [31]. Different types of FPs, with different sizes, were calculated and explored: 166 MACCS (MACCS keys), 307 Substructure (presence and count of SMARTS patterns for Laggner functional group classification—Sub and SubC, respectively), 881 PubChem FPs (ftp://ftp.ncbi.nlm.nih.gov/pubchem/specifications/pubchem_fingerprints.txt), 1024 CDK (circular fingerprints), and 1024 CDK extended (Ext circular fingerprints with additional bits describing ring features). The 1D&2D molecular descriptors comprise 1443 descriptors, including electronic, topological, and constitutional descriptors.

### 3.3. Selection of Descriptors and Optimization of QSAR Models

In the quest for QSAR models with the minimum possible number of descriptors, descriptor selection was performed based on the importance of descriptors assessed by random forest (RF) (computeAttributeImportance) [32]. Selection of descriptors was accomplished using this procedure with the importance of descriptors assessed by RF within an OOB methodology using the 50, 100, 150, and 200 most important descriptors and RF algorithm as ML technique employing the following statistical metrics: true A (TA), true B (TB), true C (TC), false A that was B (FA_B), false A that was C (FA_C), false B that was A (FB_A), false B that was C (FB_C), false C that was A (FC_A), false C that was B (FC_B), sensitivity (SE, prediction accuracy for class A, active against SARS-CoV-2), specificity B (SP_B, prediction accuracy for class B, intermediate activity against SARS-CoV-2), specificity C (SP_C, prediction accuracy for class C, inactive against SARS-CoV-2), overall predictive accuracy (Q), and Matthews correlation coefficient (MCC).

### 3.4. Class Balancer

In general, class imbalance introduces a bias in the performance of the ML algorithms due to their preference towards the majority class [37]. Our SARS-CoV-2 activity training set is unbalanced, and the imbalance ratio is 9:8:84 for the A: active/ B: intermediate/ C: inactive classes, respectively. To address this issue, a base-classifier with RF in R program version 3.6.1 [38], was used using the sampsize, which is a parameter in the RF with R specified for balancing the classes. This parameter was set to be of the same size as the minority class (class B). With this parameter, some molecules belonging to the minority class were used more than once.

### 3.5. Machine Learning (ML) Method

#### Random Forest (RF)

A RF [32,39] is implemented as an ensemble of unpruned classification trees which are created using bootstrap samples of the training set. For each individual tree, the best split at each node is defined using a randomly selected subset of descriptors. Each individual tree is created using a different training and validation set. Prediction is made by a majority vote of the classification trees in the forest. Performance is internally assessed with the prediction error for the objects left out in the bootstrap procedure (OOB estimation). The method quantifies the importance of a descriptor by the increase in misclassification occurring when the values of the descriptor are randomly permuted, correlated with the mean decrease in accuracy parameter. RFs also assign a probability to every prediction based on the number of votes obtained by the predicted class. RFs were grown with the R program [38], version 3.6.1, using the random forest library [40]. As a result of the nature of three-class imbalance, this problem was alleviated by setting the class weight ranges from 1–265, 1–265, and 1–265 for class A, B, and C, respectively, using the sampsize parameter.

### 3.6. Molecular Docking

A list of 494 virtual screening hits were prioritized by the QSAR approach from the three MNPs libraries (491, 1, and 2 from MNPs from Reaxys^®^ database, in-house MNPs, MNPs clinical pipeline, respectively). The software program OpenBabel (version 2.3.1) [41] was used to convert the mol2 files to PDBQT files. PDBQT files were used for docking to M^pro^ enzyme (PDB ID: 6LU7) [3] with AUTODOCK VINA (version 1.1) [33]. Water molecules and N3 ligand (https://www.tocris.com/products/mpro-n3_7230) were removed from 6LU7 [3] prior to docking using the AutoDockTools (http://mgltools.scripps.edu/). An experiment to evaluate the importance of water molecules in the binding of ligands to the active site of the protease was carried out and it was concluded that water molecules do not seem to have a relevant role, since a Pearson correlation of 0.885 between the calculated free binding energies with and without water molecules was obtained. The AutoDockTools is graphical front-end for setting up and running AutoDock—an automated docking software designed to predict how small molecules bind to a receptor of known 3D structure. The coordinates of the search space for M^pro^ enzyme were maximized to allow the entire macromolecule to be considered for docking. The search space coordinates were: M^pro^ enzyme; Center X: -36.149 Y: -3.796 Z: 45.045, and X: -12.806 Y: 18.646 Z: 65.607, Dimensions X: 40.000 Y: 40.000 Z: 40.000. Ligand tethering of the M^pro^ enzyme was performed by regulating the genetic algorithm (GA) parameters, using 10 runs of the GA criteria. The docking binding poses were visualized with PyMOL Molecular Graphics System, Version 2.0 Schrödinger, LLC.

## 4. Conclusions

The current results suggest that CADD approaches relying on ligand- and structure-based methodologies could be used with success to predict new inhibitory MNPs against SARS-CoV-2 M^pro^. Five MNPs, a benzo[f]pyrano[4,3-b]chromene (Reaxys ID 7450892), notoamide I (Reaxys ID 19384758), emindole SB beta-mannoside (Reaxys ID 26845562), and two bromoindole derivatives (Reaxys IDs 10714788 and 10720065) were proposed as the most promise marine drug-like leads as SARS-CoV-2 M^pro^ inhibitors. To our knowledge, the CADD ligand-based using a QSAR classification model, that was developed here, is the largest study ever performed with regard both to the number of molecules involved and to the number of structural families involved in the modeling of the antiviral activity against SARS-CoV-2 and the best model achieved an overall predictive accuracy of up to 67% for both test and training sets. In future works, the proposed five marine drug-like leads against SARS-CoV-2 M^pro^ can be validated experimentally. Based on these results, virtual libraries of marine-derived drug-like leads can be built, which can be virtual screened using the best QSAR model and molecular docking against SARS-CoV-2 M^pro^.

## Figures and Tables

**Figure 1 marinedrugs-18-00633-f001:**
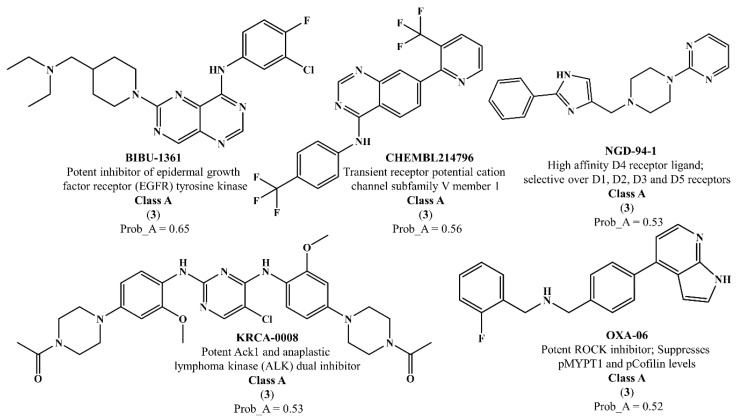
Chemical structures of the five molecules in test set 2 that were predicted as class A with the confidence value (3) and Prob_A ≥ 0.5.

**Figure 2 marinedrugs-18-00633-f002:**
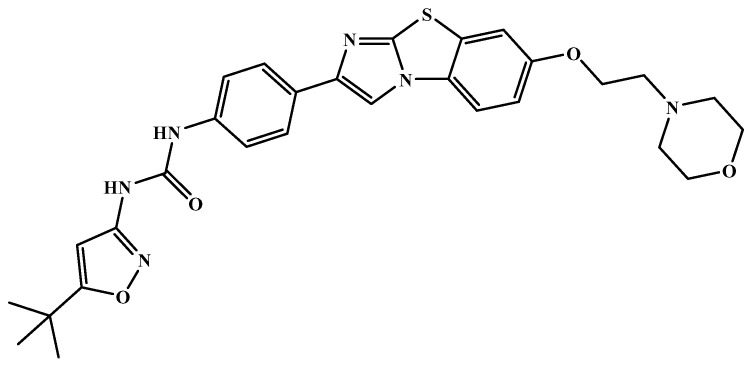
Chemical structure of quizartinib from the test set 2 that was predicted as class A with the confidence value (3) and Prob_A of 0.46, which is active in the SARS-CoV-2 CPE assay.

**Figure 3 marinedrugs-18-00633-f003:**
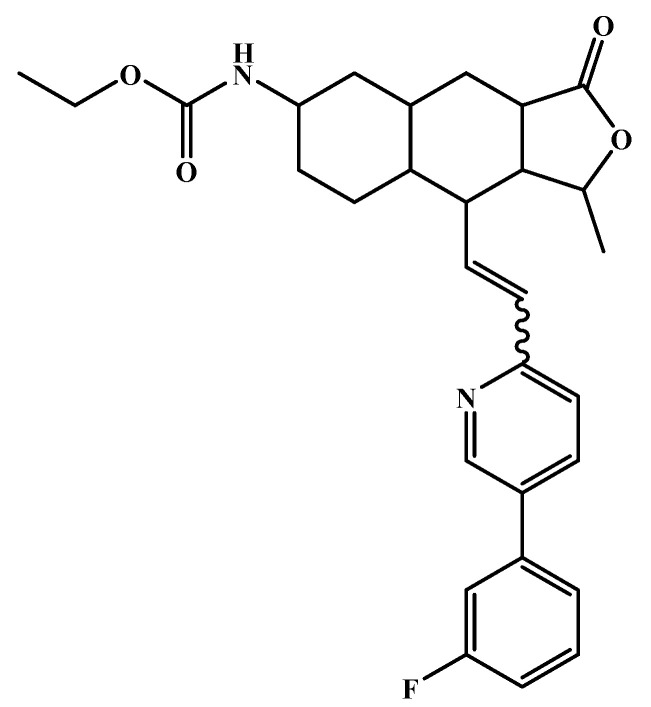
Chemical structure of vorapaxar from the test set 2 that was predicted as class A with the confidence value (1) and Prob_A of 0.17, which is active in the SARS-CoV-2 CPE assay.

**Figure 4 marinedrugs-18-00633-f004:**
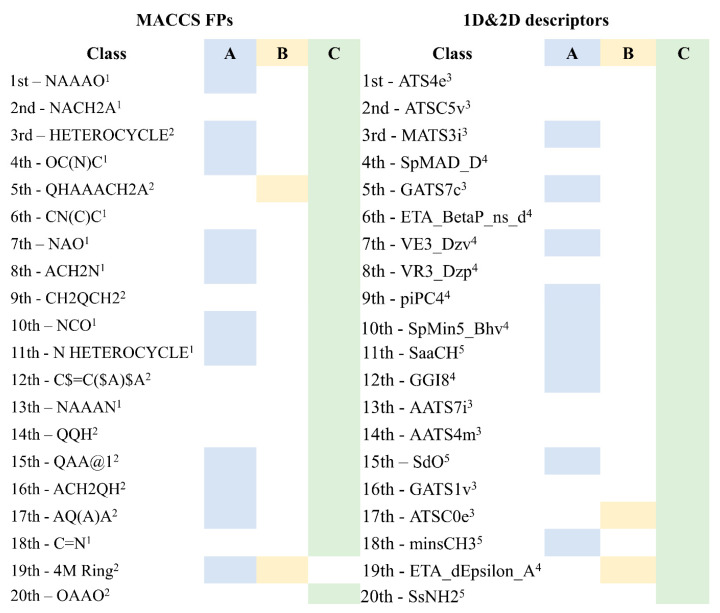
The twenty most important MACCS FPs and 1D&2D descriptors selected in RF classification models. Where: ^1^ FPs with a nitrogen atom; ^2^ FPs that can have at least one nitrogen atom as A (any atom) or as Q (any non-C or non-H atom); ^3^ Autocorrelation descriptors; ^4^ Topological descriptors; ^5^ Atom-type descriptors.

**Figure 5 marinedrugs-18-00633-f005:**
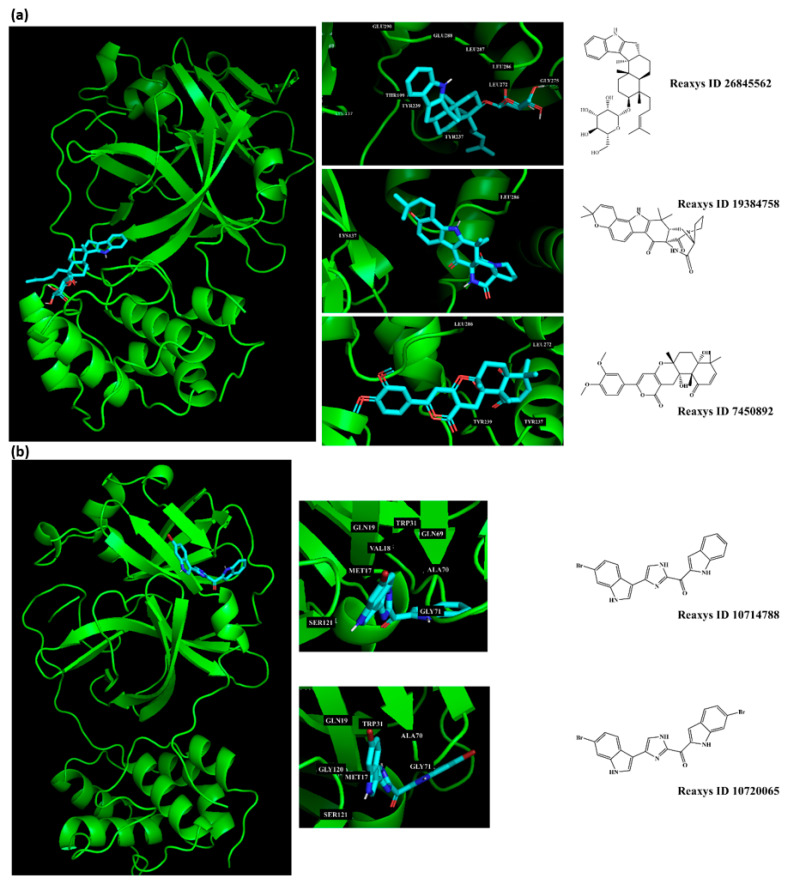
Interaction profiles of the best-docked poses for the five hits using the set search space coordinates, in which a higher affinity was accomplished (**a**) X: −36.149 Y: −3.796 Z: 45.045; or (**b**) X: −12.806 Y: 18.646 Z: 65.607.

**Table 1 marinedrugs-18-00633-t001:** Structural clusters and SARS-CoV-2 activity class counts within the ten structural clusters.

Clusters ^1^	# ^2^ (A class) ^3^	Average MW ^4^	Average ALogP ^5^
Tr Set	Te Set	Tr Set	Te Set	Tr Set	Te Set
**I**—Indole derivative 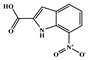	410 (41)	186 (17)	377.71	389.00	2.79	3.10
**II**—Benzoate derivative 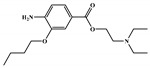	333 (35)	149 (14)	404.26	412.74	3.59	3.24
**III**—γ-Lactone derivative 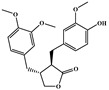	294 (10)	112 (7)	296.10	302.50	3.05	3.14
**IV**—Benzimidazole derivative 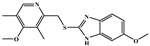	516 (57)	224 (32)	404.77	408.17	2.86	3.10
**V**—α-Amino acid ester derivative 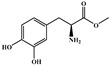	270 (23)	110 (8)	464.60	445.25	2.83	2.76
**VI**—Quinoline derivative 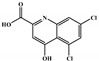	258 (28)	132 (9)	400.27	412.95	2.92	2.63
**VII**—Piperidine derivative 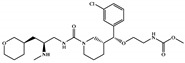	279 (13)	131 (13)	383.15	355.98	3.07	2.81
**VIII**—Acyclic derivative 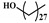	588 (31)	254 (20)	329.16	336.52	1.89	1.63
**IX**—Oxazole derivative 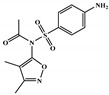	285 (29)	136 (13)	422.11	397.43	3.62	3.75
**X**—Piperidine derivative 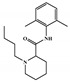	266 (35)	98 (12)	395.85	389.66	3.68	3.61

^1^ Cluster code and chemical structure of the cluster centroid. ^2^ Number of molecules in the training (Tr) and the test (Te) sets. ^3^ Number of molecules belonging to the class A in the training (Tr) and the test (Te) sets.^4^ Molecular weight (MW) within the cluster for the training and test sets. ^5^ Octanol-water partition coefficient prediction within the cluster for the training and test sets.

**Table 2 marinedrugs-18-00633-t002:** Evaluation of the predictive performance of FPs and 1D&2D molecular descriptors for modeling the antiviral activity against SARS-CoV-2 of organic molecules using the RF algorithm for the training set in an OOB estimation.

Model	MACCS	Sub	SubC	PubChem	CDK	CDKExt	1D&2D
TA ^1^	156	185	193	164	132	131	164
TB ^2^	43	64	49	37	33	37	24
TC ^3^	2017	1183	1574	1960	2170	2208	2100
FA_B ^4^	57	94	76	58	53	51	55
FA_C ^5^	567	1029	827	568	508	494	646
FB_A ^6^	29	38	37	34	17	19	14
FB_C ^7^	351	723	534	407	257	233	186
FC_A ^8^	117	79	72	104	153	152	124
FC_B ^9^	165	107	140	170	179	177	186
SE ^10^	0.52	0.61	0.64	0.54	0.44	0.43	0.54
SP_B ^11^	0.16	0.24	0.18	0.14	0.12	0.14	0.09
SP_C ^12^	0.69	0.40	0.54	0.67	0.74	0.75	0.72
Q ^13^	**0.63**	0.41	0.52	0.62	0.67	**0.68**	**0.65**
MCC ^14^	**0.27**	0.20	0.25	0.25	0.26	**0.27**	**0.26**

^1^ True A. ^2^ True B. ^3^ True C. ^4^ False A that was B. ^5^ False A that was C. ^6^ False B that was A. ^7^ False B that was C. ^8^ False C that was A. ^9^ False C that was B. ^10^ Sensitivity, the ratio of true A to the sum of true A and false A. ^11^ Specificity B, the ratio of true B to the sum of true B and false B. ^12^ Specificity C, the ratio of true C to the sum of true C and false C. ^13^ Overall predictive accuracy, the ratio of the sum of true A, true B, and true C to the sum of true A, true B, true C, false A, false B, and false C. ^14^ MCC, Matthews correlation coefficient.

**Table 3 marinedrugs-18-00633-t003:** Exploration of descriptor selection using the RF algorithm with ExtCDK FP and 1D&2D descriptors for the training set.

Model	SE ^1^	SP_B ^2^	SP_C ^3^	Q ^4^	MCC ^5^
ExtCDK FP					
50 ^6^	0.41	0.19	0.65	0.59	0.21
100 ^6^	0.44	0.14	0.70	0.64	0.23
150 ^6^	0.46	0.17	0.71	**0.64**	**0.27**
200 ^6^	0.44	0.14	0.71	0.65	0.23
1D&2D descriptors					
50 ^6^	0.54	0.11	0.69	0.63	0.25
100 ^6^	0.59	0.09	0.70	0.64	0.25
150 ^6^	0.55	0.12	0.69	**0.63**	**0.26**
200 ^6^	0.55	0.08	0.71	0.65	0.22

^1^ Sensitivity, the ratio of true A to the sum of true A and false A. ^2^ Specificity B, the ratio of true B to the sum of true B and false B. ^3^ Specificity C, the ratio of true C to the sum of true C and false C. ^4^ Overall predictive accuracy, the ratio of the sum of true A, true B, and true C to the sum of true A, true B, true C, false A, false B, and false C. ^5^ MCC, Matthews correlation coefficient. ^6^ The most important descriptors in accordance with the mean decrease in accuracy.

**Table 4 marinedrugs-18-00633-t004:** Performance of the CM predicting the antiviral activity against SARS-CoV-2 for the training and test sets.

CM	SE ^1^	SP_B ^2^	SP_C ^3^	Q ^4^	MCC ^5^
Training set ^6^	0.51	0.14	0.74	0.68	0.31
Test Set	0.48	0.08	0.74	0.67	0.19

^1^ Sensitivity, the ratio of true A to the sum of true A and false A. ^2^ Specificity B, the ratio of true B to the sum of true B and false B. ^3^ Specificity C, the ratio of true C to the sum of true C and false C. ^4^ Overall predictive accuracy, the ratio of the sum of true A, true B, and true C to the sum of true A, true B, true C, false A, false B, and false C. ^5^ MCC, Matthews correlation coefficient. ^6^ OOB estimation.

**Table 5 marinedrugs-18-00633-t005:** The predictions of the best CM model by the ten structural clusters for the test set. The best models are highlighted in bold.

Clusters	#	SE ^1^	SP_B ^2^	SP_C ^3^	Q ^4^	MCC ^5^
**I**	186	0.71	---	0.73	**0.68**	**0.32**
**II**	149	0.36	0.13	0.75	**0.68**	**0.29**
**III**	112	---	---	0.81	**0.71**	**0.37**
**IV**	224	0.69	---	0.58	0.55	0.21
**V**	110	0.25	0.20	0.74	0.68	0.05
**VI**	132	0.67	0.29	0.70	**0.67**	**0.38**
**VII**	131	0.38	---	0.82	**0.74**	**0.25**
**VIII**	254	0.30	0.11	0.81	**0.72**	**0.25**
**IX**	136	0.38	---	0.79	0.66	0.13
**X**	98	0.50	0.40	0.64	0.61	0.21
All	1533	0.48	0.08	0.74	0.67	0.19

^1^ Sensitivity, the ratio of true A to the sum of true A and false A. ^2^ Specificity B, the ratio of true B to the sum of true B and false B. ^3^ Specificity C, the ratio of true C to the sum of true C and false C. ^4^ Overall predictive accuracy, the ratio of the sum of true A, true B, and true C to the sum of true A, true B, true C, false A, false B, and false C. ^5^ MCC, Matthews correlation coefficient.

**Table 6 marinedrugs-18-00633-t006:** Structures and calculated free binding energies (∆G_B_, in kcal/mol) of the fifteen selected MNPs, one from in-house MNPs (hydroxydebromomarinone), two from MNPs pharmaceutical pipeline (nelarabine and fludarabine), and the positive (nelfinavir and lopinavir) and negative (allicin) controls, using two sets of search space coordinates.

Code	Chemical Structure	Structural Category	Natural Source	Prob_A	∆G_B_ (kcal/mol)
22947654 ^1^	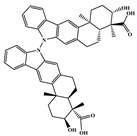	carbazole	marine derived bacteria	0.42	−9.9 ^6^/−7.6 ^7^
22947655 ^1^	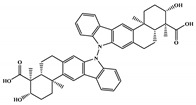	carbazole	marine derived bacteria	0.42	−9.9 ^6^/−7.6 ^7^
22435742 ^1^	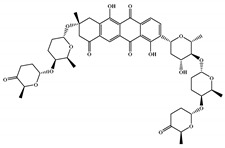	anthraquinone	marine derived bacteria	0.42	−9.4 ^6^/−7.8 ^7^
22435744 ^1^	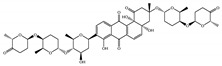	anthraquinone	marine derived bacteria	0.41	−9.4 ^6^/−7.8 ^7^
30380251 ^1^	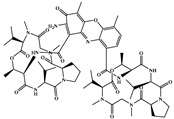	phenoxazinone	marine derived bacteria	0.68	−9.1 ^6^/−6.9 ^7^
19600610 ^1^	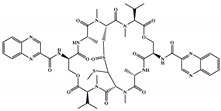	quinoxaline	marine derived bacteria	0.62	−8.9 ^6^/−8.9 ^7^
22435741 ^1^	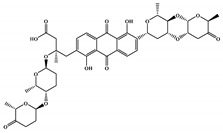	anthraquinone	marine derived bacteria	0.40	−8.8 ^6^/−7.8 ^7^
7450892 ^1^	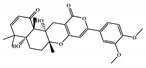	benzo[f]pyrano[4,3-b]chromene	marine derived fungus	0.41	−8.4 ^6^/−6.9 ^7^
19384758 ^1^	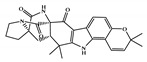	prenylated indole alkaloids	marine derived fungus	0.40	−8.4 ^6^/−7.4 ^7^
26845562 ^1^	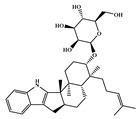	indoloditerpenes	marine derived fungus	0.41	−8.2 ^6^/−6.9 ^7^
19384759 ^1^	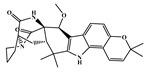	prenylated indole alkaloids	marine derived fungus	0.39	−8.1 ^6^/−7.3 ^7^
22435737 ^1^	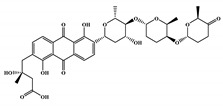	anthraquinone	marine derived bacteria	0.41	−8.0 ^6^/−7.0 ^7^
30380253 ^1^	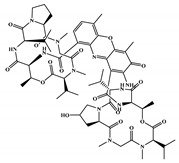	phenoxazinone	marine derived bacteria	0.59	−8.0 ^6^/−8.5 ^7^
10714788 ^1^	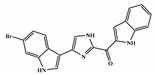	bromo deoxytopsentin	sponge	0.38	−7.6 ^6^/−8.3 ^7^
10720065 ^1^	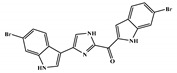	dibromodeoxytopsentin	sponge	0.38	−7.6 ^6^/−8.5 ^7^
PTM346F6F45 ^2^	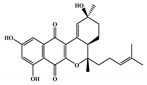	marinone	marine derived bacteria	0.30	−7.0 ^6^/−5.5 ^7^
nelarabine (Arranon^®^) ^3^	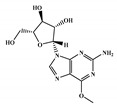	purine	sponge	0.31	−5.4 ^6^/−5.5 ^7^
fludarabine phosphate (Fludara^®^) ^3^	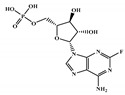	purine	sponge	0.31	−5.8 ^6^/−6.5 ^7^
nelfinavir ^4^	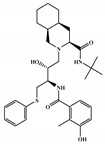	octahydro 1H-isoquinoline	---	---	−7.4 ^6^/−6.7 ^7^
lopinavir ^4^	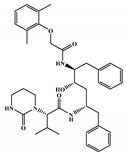	2-oxotetrahydro pyrimidine	---	---	−6.5 ^6^/−6.0 ^7^
allicin ^5^	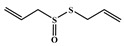	diallyl thiosulfinate	---	---	−3.3 ^6^/−2.9 ^7^

^1^ Reaxys ID from the fifteen selected MNPs. ^2^ In-house MNPs library. ^3^ MNPs clinical pipeline library. ^4^ Positive controls. ^5^ Negative control. ^6^ M^pro^ enzyme: center X: −36.149 Y: −3.796 Z: 45.045. ^7^ M^pro^ enzyme: center X: −12.806 Y: 18.646 Z: 65.607.

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
