# Peer review of "A Computer-Aided Drug Design Approach to Predict Marine Drug-Like Leads for SARS-CoV-2 Main Protease Inhibition"

_marinedrugs, 2020, doi:10.3390/md18120633_

Round 1

Reviewer 1 Report

Manuscript title: A computer-aided drug design approach to predict marine drug-like leads for SARS-CoV-2 main protease inhibition

There is an urgent need to develop therapeutic options to fight the outbreak of a novel Coronavirus (SARS-CoV-2). Computer-aided drug design (CADD) approaches can significantly accelerate the identification of drug candidates suitable for clinical evaluation. In this study, a list of virtual screening hits comprising fifteen MNPs has assented based on some established limits. The results are attempted to be rationalized by molecular docking studies.

In my opinion, the manuscript is well written, logically laid-out, and the technical content is adequate to merit the publication. The authors are urged to consider the following comments.

  1. Keep the only specific keywords
  2. Delete the methodology part from the abstract and include only major findings.
  3. Please cite the similar experimental/computational studies involving SARS-CoV-2 main protease inhibitors in the introduction or discussion.

Fu, L., Ye, F., Feng, Y. et al. Both Boceprevir and GC376 efficaciously inhibit SARS-CoV-2 by targeting its main protease. Nat Commun 11, 4417 (2020). https://doi.org/10.1038/s41467-020-18233-x

Amin O. Elzupir. Inhibition of SARS-CoV-2 main protease 3CLpro by means of α-ketoamide and pyridone-containing pharmaceuticals using in silico molecular docking. Journal of Molecular Structure, 1222, 128878 (2020), https://doi.org/10.1016/j.molstruc.2020.128878

El-hoshoudy, A. N. “Investigating the Potential Antiviral Activity Drugs against SARS-CoV-2 by Molecular Docking Simulation.” Journal of Molecular Liquids 318 (2020): 113968. https://doi.org/10.1016/j.molliq.2020.113968.

  1. The introduction can be strengthened by clarifying the novelty of the study.
  2. In conclusion, it would be very informative for the readers to include more specific drug design considerations that are a result of this study.

Author Response

1. Keep the only specific keywords. 

The “Computer-Aided Drug Design (CADD)” keyword was removed from the keyword list.

2. Delete the methodology part from the abstract and include only major findings.

The following sentences were deleted:

- “The antiviral activity was set up through three activity classes: (A) - inhibition % ≥ 50%; (B) – 50% > inhibition % ≥ 30%; (C) - inhibition % < 30%.”;

- “…comprising 1,536 and 3,499 organic molecules, respectively.”;

- “…such as: confidence value (3), probability of being active against SARS-CoV-2 in the best model, prediction of the affinity between the Mpro of the selected MNPs through molecular docking.

The following sentence was added:

“…and five MNPs were proposed as the most promise marine drug-like leads as SARS-CoV-2 Mpro inhibitors, a benzo[f]pyrano[4,3-b]chromene, notoamide I, emindole SB beta-mannoside, and two bromoindole derivatives.”

3. Please cite the similar experimental/computational studies involving SARS-CoV-2 main protease inhibitors in the introduction or discussion.

Fu, L., Ye, F., Feng, Y. et al. Both Boceprevir and GC376 efficaciously inhibit SARS-CoV-2 by targeting its main protease. Nat Commun 11, 4417 (2020). https://doi.org/10.1038/s41467-020-18233-x

Amin O. Elzupir. Inhibition of SARS-CoV-2 main protease 3CLpro by means of α-ketoamide and pyridone-containing pharmaceuticals using in silico molecular docking. Journal of Molecular Structure, 1222, 128878 (2020), https://doi.org/10.1016/j.molstruc.2020.128878

El-hoshoudy, A. N. “Investigating the Potential Antiviral Activity Drugs against SARS-CoV-2 by Molecular Docking Simulation.” Journal of Molecular Liquids 318 (2020): 113968. https://doi.org/10.1016/j.molliq.2020.113968.

4. The introduction can be strengthened by clarifying the novelty of the study.

Introduction, lines 100-103, the following sentences was added:

“Five MNPs, a benzo[f]pyrano[4,3-b]chromene, notoamide I, emindole SB beta-mannoside, and two bromoindole derivatives were proposed as the most promise marine drug-like leads as SARS-CoV-2 Mpro inhibitors.”

5. In conclusion, it would be very informative for the readers to include more specific drug design considerations that are a result of this study.

Conclusion lines 475-479, a new sentence was added.

“In future works, the proposed five marine drug-like leads against SARS-CoV-2 Mpro can be validated experimentally. Based on these results, virtual libraries of marine-derived drug-like leads can be built, which can be virtual screened using the best QSAR model and molecular docking against SARS-CoV-2 Mpro.”

Reviewer 2 Report

The manuscript details the use of CADD approach to identify potential anti-viral MNPs as inhibitors of SARS-CoV-2. 

Here are a few points to consider.

  1. Abstract can be precise with the objective and the major findings
  2. Add details about the number of GA runs set for AUTODOCK VINA runs.
  3. Few typos like “AUTODOC VINA” should be “AUTODOCK VINA”
  4. Conclusion section can be improvised by iterating your significant findings and further experiments that can be utilized to validate your findings.

Author Response

1. Abstract can be precise with the objective and the major findings.

The abstract was changed accordingly.

The following sentences were deleted:

- “The antiviral activity was set up through three activity classes: (A) - inhibition % ≥ 50%; (B) – 50% > inhibition % ≥ 30%; (C) - inhibition % < 30%.”;

- “…comprising 1,536 and 3,499 organic molecules, respectively.”;

- “…such as: confidence value (3), probability of being active against SARS-CoV-2 in the best model, prediction of the affinity between the Mpro of the selected MNPs through molecular docking.

The following sentence was added:

“…and five MNPs were proposed as the most promise marine drug-like leads as SARS-CoV-2 Mpro inhibitors, a benzo[f]pyrano[4,3-b]chromene, notoamide I, emindole SB beta-mannoside, and two bromoindole derivatives.”

2. Add details about the number of GA runs set for AUTODOCK VINA runs.

Materials and Methods, lines 460-461, the following sentences was added:

“Ligand tethering of the Mpro enzyme was performed by regulating the genetic algorithm (GA) parameters, using 10 runs of the GA criteria.”

3. Few typos like “AUTODOC VINA” should be “AUTODOCK VINA”

The manuscript was changed accordingly (page 15 line 453).

4. Conclusion section can be improvised by iterating your significant findings and further experiments that can be utilized to validate your findings.

Conclusion lines 475-479, a new sentence was added.

“In future works, the proposed five marine drug-like leads against SARS-CoV-2 Mpro can be validated experimentally. Based on these results, virtual libraries of marine-derived drug-like leads can be built, which can be virtual screened using the best QSAR model and molecular docking against SARS-CoV-2 Mpro.”

Reviewer 3 Report

Gaudencio et al. presents in this manuscript a computer-aided study aiming to the discovery of natural products obtained by marine agents acting as SARS-CoV-2 Mpro inhibitors. This research is very interesting but moreover highly original since very few (<100) hits are recovered form PubMed on this nowadays hot topic.  

From a computational point of view the article is well written, data are achieved with robust protocol especially on the QSAR modelling where the chemical space is accurately tailored from the ChEMBL, and assembled in the training and prediction sets according also to drug like criteria as defined by the Lipinski Ro5. In addition to this instance, virtual screening based on molecular docking finally resulted in a good achievement like the selection of five plausible candidates, with different chemical scaffold, that might be potentially considered as antiviral agents.

Therefore, the paper might be accepted with only minor interventions or additions, as follow:

  1. Authors only take into account Mpro protease as the main mark for the assembled library, whereas it might interest also to fishing others viral molecular targets (i.e. spike protein, ACE etc) that are known to be essential for coronavirus infections and might indeed be considered
  2. Prior to AutoDock Vina dockings waters from 6LJU7 structure were removed, whereas the same molecules might assist the binding to the protease active site. I do suggest reconsidering this aspect
  3. Some typos error must be checked

As long as this is done, in my opinion this paper might be submitted to different journal with a more pronounced pharmaceutical style

Author Response

Reviewer 3

1. Authors only take into account Mpro protease as the main mark for the assembled library, whereas it might interest also to fishing others viral molecular targets (i.e. spike protein, ACE etc) that are known to be essential for coronavirus infections and might indeed be considered.

The references 8 (introduction line 46), 9 ((introduction line 46), and 19 (introduction lines 80-81) were added.

  1. Fu, L.; Ye, F.; Feng, Y.; Yu, F.; Wang, Q.; Wu, Y.; Zhao, C.; Sun, H.; Huang, B.; Niu, P., et al. Both Boceprevir and GC376 efficaciously inhibit SARS-CoV-2 by targeting its main protease. Nat. Commun. 2020, 11, doi:10.1038/s41467-020-18233-x.
  2. El-Hoshoudy, A.N. Investigating the potential antiviral activity drugs against SARS-CoV-2 by molecular docking simulation. J. Mol. Liq. 2020, 318, 113968-113968, doi:10.1016/j.molliq.2020.113968.

Introduction lines 80-81, the following sentences was added:

 “Another study correlated the activity against SARS-CoV-2 Mpro with the presence of a different N-heterocyclic scaffold, such as a pyridone ring [19].”

  1. Elzupir, A.O. Inhibition of SARS-CoV-2 main protease 3CLpro by means of alpha-ketoamide and pyridone-containing pharmaceuticals using in silico molecular docking. J. Mol. Struct. 2020, 1222, 128878-128878, doi:10.1016/j.molstruc.2020.128878.

Introduction lines 50-51, the following sentences was added:

“Other approaches explore other targets such as the SARS-CoV-2 spike protein that binds to the human Angiotensin-converting enzyme 2 (ACE2) receptor [12,13].”

The references 12 (introduction line 51) and 13 ((introduction line 51) were added.

“12.        Chowdhury, R.; Boorla, V.S.; Maranas, C.D. Computational biophysical characterization of the SARS-CoV-2 spike protein binding with the ACE2 receptor and implications for infectivity. Comput. Struct. Biotechnol. J. 2020, 18, 2573-2582, doi:10.1016/j.csbj.2020.09.019.

  1. Huang, Y.; Yang, C.; Xu, X.-f.; Xu, W.; Liu, S.-W. Structural and functional properties of SARS-CoV-2 spike protein: potential antivirus drug development for COVID-19. Acta Pharmacol. Sin. 2020, 41, 1141-1149, doi:10.1038/s41401-020-0485-4.“

2. Some typos error must be checked.

The manuscript was changed accordingly (page 15 line 453).

3. Prior to AutoDock Vina dockings waters from 6LJU7 structure were removed, whereas the same molecules might assist the binding to the protease active site. I do suggest reconsidering this aspect.

To address this concern, the virtual screening of the 494 MNPs from the three MNPs libraries and the positive (nelfinavir and lopinavir) and the negative (allicin) controls was done by molecular docking against Mpro enzyme 6LU7 with water molecules. The prediction of the affinity between the Mpro and the selected MNPs through molecular docking without water molecules and with water molecules (Mpro enzyme: center X: -36.149 Y: -3.796 Z: 45.045) has a Pearson correlation of 0.885. Therefore, we concluded that water molecules do not play a relevant role in the binding of these molecules to the protease active site.

Materials and Methods, lines 455-459, the following sentence was added.

“An experiment to evaluate the importance of water molecules in the binding of ligands to the active site of the protease was carried out and it was concluded that water molecules do not seem to have a relevant role, since a Pearson correlation of 0.885 between the calculated free binding energies with and without water molecules was obtained.”